# Analytical Challenges in Novel Pentavalent Meningococcal Conjugate Vaccine (A, C, Y, W, X)

**DOI:** 10.3390/vaccines12111227

**Published:** 2024-10-29

**Authors:** Pankaj Sharma, Sameer Kale, Swapnil Phugare, Sunil Kumar Goel, Sunil Gairola

**Affiliations:** Serum Institute of India Pvt Ltd., Pune 411 028, India; pankaj.sharma@seruminstitute.com (P.S.); sameer.kale@seruminstitute.com (S.K.); swapnil.phugare@seruminstitute.com (S.P.); sunil.goel@seruminstitute.com (S.K.G.)

**Keywords:** MenFive, analytics, vaccine, proteins, meningococcal, multivalent

## Abstract

Multivalent meningococcal conjugate vaccines are a significant focus for the scientific community in light of the WHO’s mission to defeat meningitidis by 2030. Well-known meningococcal vaccines such as MenAfriVac, Nimenrix, Menveo, and MenQuadfi are licensed in various parts of the world and have been successful. Recently, the World Health Organization (WHO) qualified MenFive (meningococcal A, C, Y, W, and X) conjugate vaccine, further enhancing the battery of vaccines against meningitis. The antigenic nature of the current and new serogroups, the selection of carrier proteins, and the optimal formulation of these biomolecules are pivotal parameters for determining whether a biological preparation qualifies as a vaccine candidate. Creating appropriate quality control analytical tools for a complex biological formulation is challenging. A scoping review aims to identify the main challenges and gaps in analyzing multivalent vaccines, especially in the case of novel serogroups, such as X, as the limited literature addresses these analytical challenges. In summary, the similarities in polysaccharide backbones between meningococcal serogroups (C, Y, W sharing a sialic acid backbone and A, X sharing a phosphorous backbone) along with various conjugation chemistries (such as CNBr activation, reductive amination, CDAP, CPIP, thioether bond formation, N-hydroxy succinimide activation, and carbodiimide-mediated coupling) resulting into a wide variety of polysaccharide -protein conjugates. The challenge in analyzing carrier proteins used in conjugation (such as diphtheria toxoid, tetanus toxoid, CRM diphtheria protein, and recombinant CRM) is assessing their purity (whether they are monomeric or polymeric in nature as well as their polydispersity). Additional analytical challenges include the impact of excipients, potential interference from serogroups, selection and establishment of standards, age-dependent behavior of biomolecules indicated by molecular size distributions, and process-driven variations. This article explains the analytical insights gained (polysaccharide content, free saccharide, free proteins, MSD) during the development of the MenFive vaccine and highlights the crucial gaps and challenges in testing.

## 1. Introduction

Invasive meningococcal disease (IMD) is triggered by one of the six major strains of *Neisseria meningitidis* (A, B, C, Y, W, and X) [1]. Meningitidis serogroups are geographically disseminated and show dominance accordingly. Before rampant vaccination, Men B and Men C were predominant in the US and Europe, while Men A was found in Sub-Saharan Africa. Substantial cases of Men Y have been reported in Canada, South America, and Europe. Men W is found worldwide [2], while outbreaks of Men X have been reported in Africa [3,4,5,6,7]. Fatality rates of 9–12% have been reported due to invasive meningococcal disease (IMD), which can lead to rapidly progressing meningitis and septicemia [2]. IMD can lead to pneumonia, arthritis, otitis media, epiglottitis, and pericarditis [2,8]. Overall, the disease is disabling, if not fatal, despite optimal medical care and the use of appropriate antimicrobial agents. However, survivors who manage to defeat the disease are left with physical and mental consequences in most cases, such as deafness, seizures, speech impairment, limb and digit amputation, and skin scarring [9]. Therefore, vaccines against meningococcal disease are as necessary as early diagnosis and treatment because morbidity and mortality associated with meningococcal disease can be better controlled with vaccines [2,10].

Vaccines against meningococcal disease are available as polysaccharide or polysaccharide conjugate vaccines [6]. However, an effective immune response is triggered by meningococcal protein–polysaccharide conjugates A, C, Y, and W [11,12], while subcapsular protein antigens protect against meningococcal serogroup B [11]. Development of a glycoconjugate vaccine against Men X was previously reported [13]. A polysaccharide conjugate vaccine targeting meningococcal serogroup X was reported in a pentavalent formulation [14] and has been listed as a WHO-prequalified vaccine. These vaccines vary in terms of the carrier proteins conjugated to polysaccharides, for example, Men ACWY-TT (Nimenrix- Pfizer Ireland Pharmaceuticals, Ringaskiddy, Co., Cork, Ireland) and the recently licensed MenQuadfi-Sanofi Pasteur use tetanus toxoid (TT), Men ACWY-DT (Menactra-, Sanofi Pasteur Inc., Swiftwater, PA, USA) uses diphtheria toxoid (DT), and Men ACWY-CRM (Menveo-GSK, GSK Vaccines, Srl Bellaria-Rosia 53018, Sovicille (SI), Italy) uses a nontoxic mutant of diphtheria toxin (CRM197) as a carrier protein [15]. Menveo contains 10 µg of capsular oligosaccharide of serogroup A and 5 µg each of polysaccharides C, W, and Y, conjugated to 32.7–64.1 µg of the diphtheria CRM197 protein [2]. Menactra contains 4 µg of each capsular polysaccharide of serogroups A, C, W, and Y conjugated to 48 µg of diphtheria toxoid protein [2]. Nimenrix contains 5 µg of each capsular polysaccharide from serogroups A, C, W, and Y conjugated to 44 µg of tetanus toxoid protein. MenFive (A, C, Y, W, and X) vaccine contains 5 µg of each capsular polysaccharide; serogroups C, Y, and W conjugated to 15–20 µg of CRM197 protein, whereas serogroups A and X conjugated to 10–20 µg of tetanus toxoid. Polysaccharides (antigens) and carrier protein concentrations are formulated for optimized immune response. Earlier, three different carrier proteins were reported in the case of pneumococcal polysaccharide conjugate vaccine (Synflorix, GlaxoSmithKline Inc., 100 Milverton Drive, Mississauga, ON, Canada). For the meningococcal polysaccharide conjugate vaccine, two different carrier proteins were introduced and materialized for the first time in the form of MenFive. Despite the ample literature on newly developed and existing meningococcal conjugate vaccines, quality control testing and analysis demand more insight. Very few literature sources discuss the estimation of polysaccharide content in multivalent meningococcal conjugate vaccines [6,13,14,16]. Free saccharide is a vital stability-indicating parameter, requiring a thorough scientific understanding of these molecules’ separation, quantitation, and behavior. However, the available literature in this area is inadequate, and search results reflect this scarcity [14,17,18,19,20,21]. The molecular size distribution (MSD) of conjugate vaccines is another critical area of analytics that needs to be highlighted in depth. MSD analysis in monovalent conjugate vaccines or polysaccharide vaccines is well established. Still, very few reports on Hib conjugate and meningococcal C conjugate vaccines explore the application of high-performance size-exclusion chromatography (HPSEC) for MSD [22,23]. On the other side, the focus on multivalent conjugate vaccine analysis by HPSEC is minimal. Earlier reports on MSD analysis of multivalent meningococcal vaccine [24] were focused on polysaccharide vaccines. Recently, we reported the studies and behavior of multivalent conjugate vaccines through MSD analysis [25]. Similarly, free protein assessment, a counterpart to the free polysaccharide, is also an indispensable analytical parameter in the context of future analytics and the quality of multivalent meningococcal conjugate vaccines. Therefore, this review aims to provide a comprehensive overview of the analytical challenges in pentavalent meningococcal vaccine (A, C, Y, W, and X) testing, with a precise focus on the unique characteristics of the meningococcal serogroup X.

## 2. *Neisseria meningitidis* and Vaccine Development

*N. meningitidis* is a human pathogen that sources significant morbidity and mortality among infants, toddlers, and young adults worldwide through meningitis and septicemia. The organism is a fastidious, encapsulated, aerobic Gram-negative diplococcus, oxidase-positive, and utmost strains utilize maltose. *N. meningitidis* exhibits epidemiological variation in its expression of capsular polysaccharides (serogroups A, B, C, W, Y, and X) or non-capsular antigens [26]. However, most reports of invasive meningococcal disease (IMD) are caused by strains that express typeable capsular polysaccharides, confirming a significant factor in pathogenesis. The virulence of *N. meningitidis* is mediated through multiple factors, including capsule polysaccharide expression, surface adhesive proteins (such as outer membrane proteins and pili), iron sequestration mechanisms, and endotoxin lipo-oligosaccharide (LOS) [27]. In the early 1900s, outbreaks of serogroup A were common in the United States. However, after the 1950s, *N. meningitidis* A disappeared from the US and other industrialized countries for unknown reasons [26]. Despite the licensing of a quadrivalent conjugate vaccine (A, C, Y, and W) in the USA [28,29,30,31,32], frequent cases of serogroup C, Y, and W continued to appear, possibly due to low vaccine uptake rates of 11.4% in 2006 and 32.4% in 2007 [26,33]. On the other side, Europe and the UK have reported comparatively more cases of *N. meningitidis* than the USA, with 2–5 cases per 100,000 per year [26]. In 1999, the first meningococcal conjugate vaccine targeting serogroup C was introduced in the UK, resulting in a decreased incidence of serogroup C disease. The term “Meningitis Belt” gained worldwide recognition when Lapeyssonnie accredited a region in African countries [34] experiencing periodic epidemics and endemic meningococcal disease, primarily of serogroup A, which continued until the introduction of the conjugate vaccine MenAfrivac targeting serogroup A in 2010 [35]. Among the six deadliest serogroups, those above five (A, C, Y, W, and B) are well known worldwide for their spread, preventive tactics, available vaccines, and immunogenicity. Recent outbreaks of serogroup X in African countries such as Niger, Kenya, and Ghana have raised concerns among health authorities about the severity of the disease caused by serogroup X [3,4,5,6,7]. The newly developed pentavalent vaccine containing serogroup X has shown promising results in preclinical and clinical studies. It is poised to address the infection caused by serogroup X soon [14]. WHO-prequalified vaccines are listed in Table 1.

Vaccines are complex biopharmaceutical formulations; even after 400 years since the first official vaccination report, we are still striving to understand the mechanism of action of vaccines. Vaccines are diverse biological formulations with various components, such as polysaccharide vaccines (subunit vaccines), live attenuated vaccines, killed vaccines, recombinant vaccines, viral vaccines, and many more; however, in most cases, they could be more well defined [36]. Challenges related to vaccine development are associated with multiple factors. These factors comprise understanding the complexity of proposed vaccines in terms of their antigenic components, structure, mechanism of action in the human body, ability to trigger the immune system, long-term effects, and other aspects decisive for developing a promising vaccine candidate (refer to Figure 1).

In addition to having a scientific understanding of the process, developers and manufacturers need to consider the economic and regulatory aspects of the product. One of the noteworthy challenges for any manufacturer is to warrant that the product remains affordable while also being safe and effective. The development team must ensure that the pilot-scale product, including in-process, formulated, and finished products, remains steady in behavior during manufacturing and all over its shelf life. Therefore, quality control and analytics are indispensable components of vaccine development from the very beginning. With proper analytics, it is possible to ensure the consistent and reliable delivery of the product. Assay development is considered a continuous process that evolves alongside product development and for a novel product, it is rare to have analytics in place before development. Developing an investigational new drug (IND) product is always challenging compared to established licensed products (refer to Figure 2).

The pentavalent (A, C, Y, W, and X) meningococcal conjugate vaccine being discussed was developed by Serum Institute of India Pvt. Ltd., 212/2, Off Soli Poonawalla Road, Pune-411028, India. Until now, only quadrivalent meningococcal vaccines have been available on the market either in a single formulation [28,29,30,31,32] or in a combination of different polysaccharides. A new pentavalent conjugate vaccine comprising serogroups ABCYW is also reported to be in its late clinical stages [37]. The MenFive vaccine development involved exploring different combinations of fermentation conditions, carrier proteins, antigen concentrations, polysaccharide–protein ratios for conjugation, and conjugation chemistries. The manufacturing processes for the novel X-containing vaccine formulations are summarized in an earlier publication on process development for meningococcal serogroups X [7]. Various combinations of polysaccharide–carrier proteins, including TT, CRM, and DT, were explored. Among all the combinations, the top five were selected for the final formulation based on their ability to elicit an immune response. The preferred formulations contained two conjugates with TT (A and X serogroups attached to TT) and three conjugates with CRM (C, Y, and W attached to CRM). Several challenges were noted in upholding these formulations in their optimal state, including difficulties with respect to excipient concentrations, the effect of carrier proteins, and their aggregation behavior. These aggregations not only distress protein molecules but also influence other molecules present in the final formulations. In summary, navigating the behavior of polysaccharides with counter proteins (AX with TT, and CYW with CRM) can be challenging. It is crucial to maintain the ratio of polysaccharide to protein and safeguard that the free polysaccharide content remains within acceptable limits. However, all of these challenges were successfully addressed to deliver an efficient final product: a pentavalent meningococcal vaccine, MenFive. MenFive has completed Phase I, II, and III clinical trials and has recently been licensed and WHO-prequalified [25]. The development of this pentavalent meningococcal vaccine was aligned with WHO TRS 924, 962, and 963 [38,39,40]. However, tests and a few specifications were in-house adapted (for serogroup X) over WHO TRS due to limited resources and information about formulations containing serogroup X. These embellishments will provide appreciated insights for developing new guidelines precisely for serogroup X.

## 3. Analytical Challenges and Limitations

Vaccines are categorized as biological formulations, primarily injectables. Therefore, strict regulatory guidelines are already in place for manufacturing and testing well-established vaccines. Regulatory guidelines, such as those provided by the World Health Organization’s Technical Report Series, are available for meningococcal A and C vaccines [38,39,40,41,42,43]. Since serogroups A and C are among the earliest pathogens, vaccines were established against these serogroups in the early 1970s. Limited regulatory guidelines are available for other meningococcal pathogens such as serogroup Y and W. Most of these guidelines are still under development, and the manufacturing and testing of these vaccines mainly refers to WHO TRS for meningococcal A and C vaccines. Instead, meningococcal serogroup X is relatively new, which makes the manufacturing and testing of X-containing vaccines a challenging task, particularly in the absence of definitive guidelines.

### 3.1. Key Quality Control Analysis

The present article classifies analytical challenges related to polysaccharide content estimation, development of standards and their strategic use, estimation of free saccharide content, characterization of carrier protein, application of molecular size distribution (MSD) for the multivalent conjugate vaccine, and establishment of potential system suitability criteria. Various quality control parameters are involved in testing multivalent conjugate vaccines or investigational new drug products and are categorized according to the sample processing stage or analytical methods employed (refer to Figure 3a,b).

#### 3.1.1. Challenges and Gaps Concerning Polysaccharide Content (Potency) Estimation

Protection against meningococcal infection is primarily provided by active components in most prophylactic vaccines, specifically consisting of polysaccharides derived from the bacterial capsules of the four serogroups A, C, Y, and W135 [16] with a recent addition of serogroup X [25]. The polysaccharide component is the core of the meningococcal conjugate vaccine (Table 2), making its precise estimation through suitable quality control tests of utmost importance. Various regulatory guidelines such as IP, BP, and WHO provide thorough protocols for testing the polysaccharide content in meningococcal vaccines. Meningococcal serogroups A and X are characterized by a phosphorous backbone, Men A polysaccharide is a homopolymer of α-(1→6) linked N-acetylated mannosamine-6 phosphate [16], and Men X polysaccharide is a homopolymer of α-(1→4) linked 2-acetamido-2-deoxyglucosyl-phosphate [44]. Both of these can be accurately estimated using the colorimetric Chen’s method. Aimed at the Men X serogroup, in-house specifications were established and grounded on guidelines available for other meningococcal serogroups. The phosphorus content of purified polysaccharides from Men A and Men X was confirmed to be above 8% of the dry weight of the isolated product. Similarly, serogroups C, Y, and W, which represent homopolymers of α-2,9-Neu5Ac monosaccharide, α2,6-Glc- α1,4-Neu5Ac disaccharide, and α2,6-Gal-α-1,4-Neu5Ac disaccharide, respectively, ref. [16] were tested for sialic acid content using the Resorcinol–HCl method (Svennerholm’s method). The sialic acid requirement for Men C was 80% of the dry weight of the isolated product, while Men Y and W should have no less than 560 mg/g of polysaccharide.

As mentioned earlier, testing of monovalent polysaccharides (A, C, Y, W) was well supported and guided by the previously available literature. However, the real challenge for testing began when these polysaccharides were formulated into a multivalent mixture. Chemical assays were not appropriate for quantifying multivalent vaccines as serogroups A and X share similarities in terms of phosphorus content in the backbone [16], making it unsuitable to be estimated by Chen’s method. Likewise, serogroups C, Y, and W demonstrate similarities in terms of N-acetylneuraminic acid content [16,45], restraining the use of Svennerholm’s method. Capillary zone electrophoresis (CZE) and high-performance anion-exchange chromatography with pulsed amperometric detection (HPAEC-PAD) are the preferred methods for quantifying meningococcal polysaccharides in multivalent vaccines [6,16,46]. However, CZE needs to be thoroughly explored and evaluated for the quantitation of new serogroups manufactured using new conjugation chemistries [47]. Hence, HPAEC-PAD remains the most widely accepted method for quantitating multivalent vaccines [16,21]. Being a selective quantitation method, it is also famous for its ability to quantify samples even at the picomole level [48]. In addition to these well-established methods, immunoassays like enzyme-linked immunosorbent assay (ELISA) can also be a good substitute for quantitating these multivalent vaccines [6]. In our preceding study, we explored the application of sandwich ELISA for quantifying pentavalent meningococcal conjugate vaccine in detail. The study verified adequate precision, accuracy, and robustness [6]. Despite their potential, bioassays are a less preferred choice in modern-day analytics owing to issues with lot-to-lot consistency of antibodies, standards, and other chemicals used in these assays. Therefore, we focused our analytics on the application of HPAEC-PAD for quantifying multivalent meningococcal conjugate vaccines. Although HPAEC-PAD is the most robust and consistent method, it also has some disadvantages. It is unable to estimate the product in its native form and requires the product to be degraded by acid and heat digestion into monosaccharide units. Additionally, interference from excipients needs to be addressed, thus making this method laborious and costly. Despite its limited disadvantages, HPAEC-PAD remains the most commonly used method in vaccine analytics, particularly for glycoconjugate vaccines.

Multivalent meningococcal conjugate vaccines can be effectively analyzed using HPAEC-PAD [16,21]. However, there is a condition that must be carefully considered to ensure unbiased results. Characteristic monosaccharides obtained from the hydrolysis of multivalent meningococcal polysaccharide conjugate vaccines [13,16] are outlined in Table 2. A closer look at Table 2 provides insight into the similarities between meningococcal serogroup backbones, which can complicate both chemical estimation and HPAEC-PAD analysis. Cook et al. (2013) previously reported the HPAEC-PAD method for quantifying serogroups A, C, Y, and W in a quadrivalent conjugate vaccine. They developed serogroup-specific hydrolysis conditions but noted cross-interference of N-acetylneuraminic acid among serogroups C, Y, and W, which is a point of concern. Until now, none of the formulations of multivalent meningococcal conjugate vaccines have included serogroup X. Therefore, limited information on the interference properties of X’s characteristic monosaccharide was available until our recent publication [21]. We observed that meningococcal serogroup X contributes non-specifically to meningococcal serogroup A, leading to falsely elevated levels of serogroup A unless properly addressed. While one conventional option is to correct non-specific interference by applying correction factors, we opted for a strategic standard preparation and usage application to report unbiased and accurate polysaccharide content.

In addition to counteracting serogroups, non-specific interference in serogroup Y can also be expected from the excipients. Added sugars such as sucrose or lactose in excipient formulations [16] could interfere with the estimation of serogroup Y and, in some cases, serogroup W as well. This is because the characteristic molecules for these serogroups are glucose and galactose, respectively. To address this issue, an ultrafiltration approach was adapted, which employed 3 kDa filters from various vendors (Pall, Amicon, Sartorius) to eliminate excipients before analysis.

Column selection for HPAEC-PAD analysis was crucial. Precisely, Carbopac columns were used to estimate polysaccharides or free polysaccharides. The PA1 column was used to estimate serogroup C, while serogroups A, Y, W, and X were estimated using the PA10 column. Firstly, all serogroups were tested with the PA1 column. However, switching to the PA10 column for serogroups A, Y, W, and X resulted in improved peak behavior and elution patterns over time. The literature and manufacturer’s technical notes indicated that the PA1 column had an oxygen dip issue, likely caused by dissolved oxygen in the samples. This issue was affecting the estimation of Men W (galactose) and Men Y (glucose). The peak of Men A also appeared broader on the PA1 column compared to the sharp peak on the PA10 column. Based on these verdicts, the analysis of serogroups A, Y, W, and X was shifted to the PA10 column. The elution gradient was another critical factor in ensuring optimal peak resolution for monovalent samples within a short timeframe. Therefore, gradient programs were optimized to expedite the manufacturing process. Retention times with PA1 columns for standard and hydrolyzed product monosaccharides were as follows: neuraminic acid (approximately 7.0 min), and glucuronic acid (approximately 19.0 min); with PA10 columns, mannosamine (approximately 11.2 min), galactose (approximately 14.0 min), glucose (approximately 16.0 min), N-acetylneuraminic acid (approximately 30.8 min), glucosamine one phosphate (approximately 33.5 min), mannosamine-6-phosphate (approximately 42.5 min), and glucosamine-4-phosphate (approximately 45.2 min). Results gained with Carbopac columns displayed consistent performance, requiring minimal column regeneration and negligible wear and tear. Formerly reported pristine columns [16] showed analogous retention times for standard and hydrolyzed products. However, they experienced significant wear and tear compared to CarboPac PA1 and PA 10 columns, resulting in retention time variations over time [16].

#### 3.1.2. Standards and Its Strategic Use

Selecting a polysaccharide standard at the correct downstream processing stage was a crucial step before standard qualification. We chose purified lyophilized polysaccharide as our standard, considering its moisture content. The pentavalent meningococcal vaccine testing involves samples from various phases. Intermediate process samples were primarily monovalent, whereas formulated bulk and finished fill products were multivalent. Considering this fact, it is important to note that a standard application strategy is crucial and heavily depends on the nature of samples. Therefore, the standard application strategy should be chosen as monovalent or mixture (multivalent) standards. Since serogroups A and X share a phosphorous backbone and serogroups C, Y, and W share a sialic backbone, non-specific contributions in the case of serogroups A and C can lead to overestimations. It is more appropriate to use a multivalent standards strategy for multivalent samples. On the contrary, the use of multivalent standards may not apply to monovalent samples. It can result in falsely lower values due to non-specific contributions from standards.

The final lot comprises all five serogroups in a single formulation, which means the testing strategy differs from monovalent conjugate bulks. One can choose between a monovalent standard or a multivalent standard for estimating the final lot. However, using a monovalent standard may require running multiple sequences to assess all five serogroups’ total and free polysaccharide content, which can be time-consuming. Additionally, using monovalent standards in the final lot may result in a falsely higher content for specific serogroups (A and C) because of non-specific contributions from other serogroups (Y, W, and X). Based on developmental data, it was concluded that using a multivalent standard would be more advantageous in reducing testing timelines and providing technical compensations over non-specific contributions. Our method is advantageous as it can estimate up to four serogroups (A, Y, W, and X) in a single chromatographic run. Presently, no other manufacturer includes the X serogroup in their product, making Serum Institute the first to analyze four meningococcal serogroups in a single run. A quadrivalent standard was used to estimate the total polysaccharide content and free polysaccharides of A, Y, W, and X. Moreover, a trivalent standard was used to calculate the total and free polysaccharide content of serogroup C. The digestion and other chromatographic conditions were constant for all the five serogroups in both fill finished product and conjugate bulks. In addition to polysaccharide standards, Table 3 highlights many different standards used in the analysis of meningococcal conjugate vaccines.

#### 3.1.3. Challenges and Gaps Concerning Stability Indicating Parameter: Unbound (Free) Saccharide in Monovalent and Multivalent Formulations

In vaccines, impurities may appear in various forms, and a significant impurity that can develop over time is free saccharide and it can be either process-driven or age-driven. Process-driven free saccharide refers to unconjugated saccharide remaining from the manufacturing process, while age-driven free saccharide results from molecular changes over time that impact the stability profile. Estimating free saccharide is critical for analyzing vaccine efficacy and immunogenicity. Several well-known methods are available for estimating free polysaccharides, such as ultrafiltration, DOC precipitation, solid-phase extraction, and resin-based separations like Capto adhere and Capto Q [20,49]. The challenge lies in choosing the most suitable method for our molecules. In the early stages of development, various approaches were considered, including ultrafiltration, detergent, and chemical precipitation of proteins. Initial efforts focused on ultrafiltration, testing different filter membranes (ranging from 30 kDa to 300 kDa), elution buffers, centrifugation cycles, and varying centrifugal forces. However, soon it became clear that elution based on size was not practical due to the complex nature of our vaccine, which comprised dual carrier proteins, their aggregations, and different activation strategies. To address this issue, we pursued a size- and shape-independent tactic and found that protein precipitation using solvent or detergent was a viable option for separating free saccharides. Several detergents, including sodium dodecyl sulfate and sodium deoxycholate, were tried during the experimentation. In some cases, ammonium sulfate or acetone precipitation was also utilized. Based on statistical analysis and a literature review, the DOC precipitation method was selected for estimating free saccharides in meningococcal conjugates. Initial attempts to standardize DOC conditions for all serogroups revealed variable behavior among the meningococcal conjugates. Different derivatization strategies for the five serogroups justified the need for specific DOC conditions. After testing various concentrations of DOC and precipitation methods, three optimal conditions for DOC were established for the five serogroups. Serogroups C, Y, and W responded uniformly to the same DOC conditions [21], while serogroups A and X required different approaches for optimal results [21]. The various responses to DOC precipitation among the serogroups may be attributed to derivatization and carrier protein behavior variations. The specifications for free polysaccharides were established following WHO TRS 962 Annexure 2, WHO TRS 924 Annexure 2 for serogroups A and C, the European Pharmacopoeia, and internal specifications. Serogroups Y and W specifications were based on European Pharmacopoeia and in-house specifications. In contrast, serogroup X specifications relied solely on in-house specifications due to the absence of any compendial or other regulatory requirements.

Estimating the levels of free saccharide in the final product was one of the most challenging tasks we encountered during the product development process. The final product formulation consists of five different polysaccharides and two carrier proteins that were conjugated using unique derivatization and conjugation chemistry. Therefore, determining the free saccharide content in this complex formulation proved challenging. Fortunately, we successfully determined the levels of four serogroups (A, C, Y, and W) using the same conditions as the conjugate bulk without encountering many obstacles. The analytical methods for these serogroups were validated following ICH Q2R1 guidelines and complied with all the necessary criteria. However, the final serogroup, X, offered a significant challenge when attempting to estimate the free polysaccharide levels using the existing conditions for the conjugate bulk. During validation according to ICH guidelines, it became evident that the accuracy conditions were not meeting the required standards, with almost zero percent spike recoveries noted. After numerous attempts to optimize the recoveries by adjusting the conditions, it was determined that the free polysaccharide from serogroup Men X was being trapped by one or more biomolecules in the formulation. These biomolecules could be polysaccharides from the other four serogroups (A, C, Y, and W) or the carrier proteins TT or CRM, present in either conjugated or free form. A total of 32 formulations, including polysaccharides (monovalent, bivalent, trivalent, quadrivalent), conjugate (monovalent, bivalent, trivalent, quadrivalent), and proteins (TT and CRM), were examined to determine the origin of the X polysaccharide entrapment. Multiple analyses showed that the Men X free polysaccharide was precisely trapped in CRM and CRM-based formulations, resulting in accuracy issues. It was conjectured that CRM aggregates were responsible for trapping the X-free polysaccharide. Resolving this issue was crucial before conducting a root cause analysis. By investigating various buffer combinations (Tris buffer, citrate buffer, phosphate buffer, phosphate buffer saline, phosphate buffer with tween 80) to disperse the CRM aggregates, we found optimal buffer combination in the form of phosphate buffer with tween 80 to successfully release the entrapped free polysaccharides. With the finalized buffer and DOC conditions, we were able to resolve the problem and successfully validate the product. The Men ACYWX polysaccharide conjugate vaccine now meets high-quality standards and is positioned to capture a significant portion of the current market share for meningococcal vaccines.

#### 3.1.4. Challenges and Gaps Concerning Bioassays (ELISAs) and Their Applications

The sandwich ELISA method for estimating the polysaccharide content of pentavalent meningococcal conjugate vaccine (A, C, Y, W, and X) was earlier described in our publication [6]. We explored the sandwich ELISA application for two different approaches other than the estimation of polysaccharide content. Firstly, we used ELISA to predict the antigenicity of our product at the monovalent conjugate bulk stage utilizing the concept of the antigenic ratio. The polysaccharide content of the monovalent conjugate bulk obtained through sandwich ELISA was divided by the polysaccharide content obtained through chemical assays. The resulting ratio will be considered as the antigenic ratio. The lower the ratio, the lower the antigenicity, and vice versa. Secondly, a new insight into the possible use of sandwich ELISA as a stability-indicating test for pentavalent meningococcal vaccine testing (indirect estimation of free saccharide) was hypothesized in our earlier publication [6]. Significant variation was observed when sandwich ELISA was used to quantify the multivalent meningococcal conjugate vaccine under various experimental and environmental conditions compared to control samples. This variation was correlated with increased free saccharide levels, which could be a cause for the lower quantitation response [6]. Homogenized plain polysaccharide spikes (i.e., representative-free PS) were introduced into vaccine vials. Polysaccharide spiked samples showed altered responses when analyzed through sandwich ELISA compared to control samples. This formed the basis of our hypothesis regarding the application of sandwich ELISA for exploring free saccharides. The reduced quantitation response in ELISA may be due to competition between free polysaccharide and polysaccharide conjugate epitopes to occupy the paratopes of coated polyclonal antibodies. Free saccharides, being smaller in size, have a higher binding frequency compared to polysaccharide conjugate binding. In short, smaller free/plain saccharide epitopes attach to the paratopes of capture or coated antibodies. This can lead to possible steric hindrance for primary antibodies to access the epitopes of the attached antigen, resulting in lower polysaccharide estimation. Although the sandwich ELISA developed has some limitations, it does provide a new perspective in the field of analytics for multivalent meningococcal conjugate vaccines.

#### 3.1.5. Challenges and Gaps Concerning Carrier Protein Estimation

Similar to free polysaccharides, a counterpart of the conjugate represents the free protein, and there is a robust regulatory requirement for the estimation of free protein. Since two different carrier proteins were involved, it was clear that multiple approaches are needed for free protein estimation. To the best of our knowledge, most theories are proven based on dilution theories or by demonstrating and monitoring the conversion of individual carrier proteins and polysaccharides into the formation of conjugates (conjugation reactions). This is due to the lack of methodology to estimate such a small amount of free protein accurately. Estimation of free proteins is primarily linked to conjugation chemistries, which play a crucial role and are perhaps the most significant challenge in developing methods for estimating free proteins. Conjugates formed using chemistries such as cyanogen bromide activation, carbodiimide-mediated coupling, reductive amination, the one-step method of N-hydroxyl succinimide activation, and by the formation of thioether bonds may result in conjugates that are expected to be more significant and less stable compared to conjugates formed by CDAP/CPIP chemistries. Another challenge is presented by carrier proteins such as diphtheria toxoid, tetanus toxoid, and CRM diphtheria protein, which are polymeric. This makes it difficult to separate the free protein alongside polymeric or less stable conjugate bulks during the release and stability studies. Free protein characterization methods for multivalent conjugate vaccines are less explored, and there is limited literature available on this topic.

#### 3.1.6. Challenges and Gaps Concerning Molecular Size Distribution (MSD) in Mixed Formulations

The molecular size distribution test (MSD) is another crucial indicator of the quality of conjugates and is estimated using the SEC-MALS method. A new physicochemical characterization approach, such as determining the molecular size and weight in the novel meningococcal ACYWX polysaccharide conjugate vaccine, can aid in understanding lot-to-lot consistency. It provides information on the distribution of various molecules (such as conjugates/polysaccharides/proteins) present in terms of high-molecular-weight and low-molecular-weight molecules in the sample. It can also disclose the distribution pattern of molecules that may result from different dose presentations and formulations. Estimating the pentavalent average molecular weight and the molar mass distribution post-slicing provides an understanding of the conformation of molecules. This is performed by categorizing them as high molecular weight (HMW), low molecular weight (LMW), and aggregates, identifying the dominant population of molecules based on their molecular weight and total contribution to the sample. It also offers information on the change in molecular weight from the liquid state formulation to the final lyophilized state [25]. However, a significant limitation in exploring the molecular size distribution (MSD), especially in the case of multivalent products, was the limited availability of the literature. Earlier, Khan et al., in 2016, attempted to study MSD in polysaccharide vaccines but not in conjugate vaccines. The molecular nature of the conjugate vaccine is directly related to its immunogenicity. Therefore, molecular characterization can be a critical parameter in monitoring quality and ensuring lot-to-lot consistency. The vaccine sample under inspection was complex, carrying five serogroups and two carrier proteins and each molecule showed its own aggregation/dispersity behavior. The biggest challenge was selecting the column and buffering conditions to fractionate these molecules properly. A three-column series was used with varying fractionating capacities. The initial followed the guard column (Shodex OHPAK SB-G-6B) in the series, Shodex OHPAK SB 807 HQ, with a particle size of 35 µm and an exclusion limit of 500,000 kDa, followed by Shodex OHPAK SB 806 HQ with a particle size of 13 µm and an exclusion limit of 20,000 kDa, and G6000 PWXL with a particle size of 13 µm and an exclusion limit of 8000 kDa [25]. Multiple column combinations were explored before settling on the reported combinations above. However, the majority of issues were noted regarding the recovery of injected mass and the polydispersity index, which could be associated with the complex biological nature of the vaccine. The dn/dc value is a critical parameter for calculating molecular mass from light scattering measurements, and an accurate measurement of these values correlates to molecular mass determination [23]. Without a definitive strategy for determining the dn/dc of multivalent conjugate vaccines, we utilized previously reported dn/dc values for CRM conjugates [50]. In contrast, dn/dc values for TT conjugates were derived from Men C-TT conjugates [23]. Although adapting dn/dc values from unrelated proteins or polysaccharides could be misleading [23], the estimation of the molecular weight of all the five conjugate bulks using the newly developed method aligned with our earlier report [14], confirming the method’s suitability for the intended application.

#### 3.1.7. System Suitability (Assay Validity Criteria) Establishment

Well-established system suitability criteria for any test method, whether chromatographic or chemical, ensure the appropriateness of analytical development, qualification, and validations. Therefore, before finalizing the analytics, it is essential to consider a discussion of SST. Statistical calculations help in developing various system suitability criteria. For example, in spectrophotometric assays, one could consider the correlation coefficient, percentage standard recovery, percentage CV of duplicate standards/samples, k factor, and reference standard (international or available pure compound). In chromatographic systems, additional criteria may include the capacity factor, relative retention, resolution, theoretical plates, and tailing factor as part of system suitability testing (SST). SST for non-chromatographic methods and a few chromatographic techniques have been established using well-known traditional system suitability criteria (refer to Table 4 and Table 5) and are not discussed in detail here. The establishment of system suitability, especially for chromatographic methods, offers a variety of choices for selection. For example, much of the analytics for pentavalent meningococcal vaccines involves ion-exchange chromatographic analysis. The separation of sample molecules is based on charge-bearing functional groups through anion-exchange chromatography with pH-based elution. The separation is primarily influenced by the pH, ionic strength, and temperature. Therefore, solid support from SST is necessary to monitor the performance of such a complex and sensitive chromatographic technique. In addition to traditional HPLC, we also focused on utilizing international reference standards and internal control methods to establish SST. All five meningococcal polysaccharides in vaccine formulations were evaluated using a system that monitored international standards within a predefined range during each run. An internal standard method was used to monitor the performance of samples, standards, and international standards against system response. The internal standard method involved spiking a compound of known purity into the sample, which does not interfere with the analysis. Quantitation was determined by comparing the response ratio of the compound of interest to the internal standard with a similar reference standard preparation. The internal standard system suitability method was adopted due to the complex sample preparation (multiple steps processing and extractions) and relatively low sample content. Validated spreadsheets were designed for calculation purposes wherever applicable for controlled release.

## 4. Limitations

This proposed review may have limitations, and to make our review more manageable, we only included information that primarily focused on analytics of multivalent meningococcal vaccines, considering the key parameters. Additionally, there is minimal literature available in this specific area of research, particularly on multivalent meningococcal conjugate vaccines with new serogroups like X, which was available until September 2024.

## 5. Conclusions

The limited literature supporting analytics in glycoconjugates with evolving conjugation chemistries containing new serogroups and multiple carrier proteins in one formulation poses challenges for scientists and manufacturers dealing with investigational new drugs. This review aimed to identify gaps and gain insight into the literature regarding the quality control analysis of multivalent meningococcal conjugate vaccines. The information presented here may serve as a guiding tool for a future systematic review. This advocates extensive research is warranted to determine the quality of vaccines, which may benefit future developers and manufacturers. We have developed an approach focusing on crucial analytical areas to overcome the challenges encountered. These include estimating potency, free saccharides, free protein, and addressing potential challenges and gaps in testing. One key takeaway from this review article could be the implementation challenges for MSD in understanding lot-to-lot consistency in multivalent conjugate vaccines. Overall, the manuscript narrates the challenges encountered and the approach adopted while developing quality control methods that are used for routine batch release purposes and stability studies to determine product shelf life. The information would contribute value to developing regulatory guidance for IND polysaccharide conjugate vaccines, particularly for meningococcal A, C, Y, W, and X polysaccharide conjugate vaccines.

## Figures and Tables

**Figure 1 vaccines-12-01227-f001:**
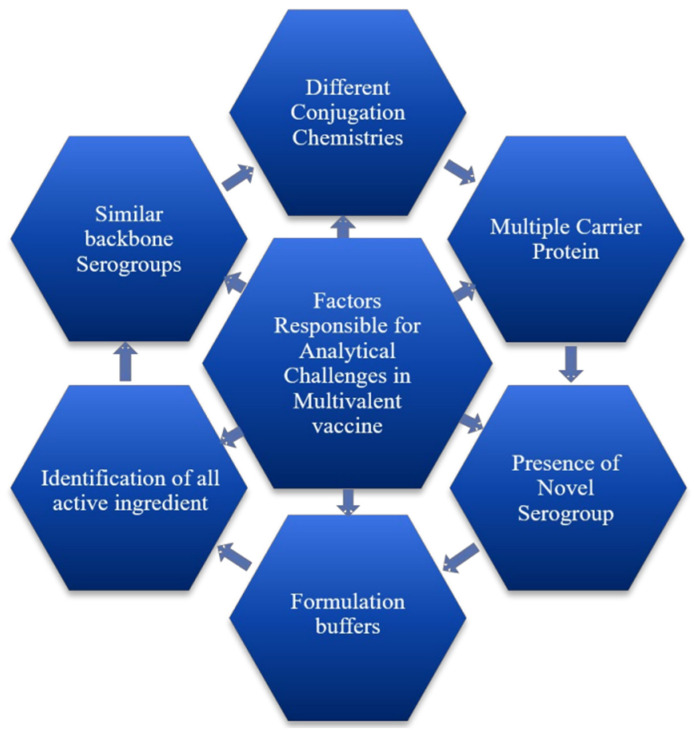
Interlinking factors accountable for analytical challenges.

**Figure 2 vaccines-12-01227-f002:**
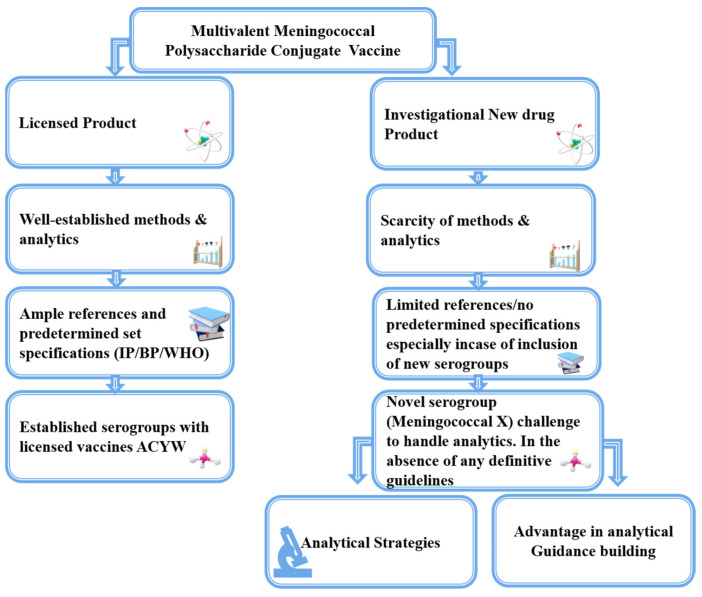
Comparative representation of analytics challenges in new vaccine analytics vs. established vaccine.

**Figure 3 vaccines-12-01227-f003:**
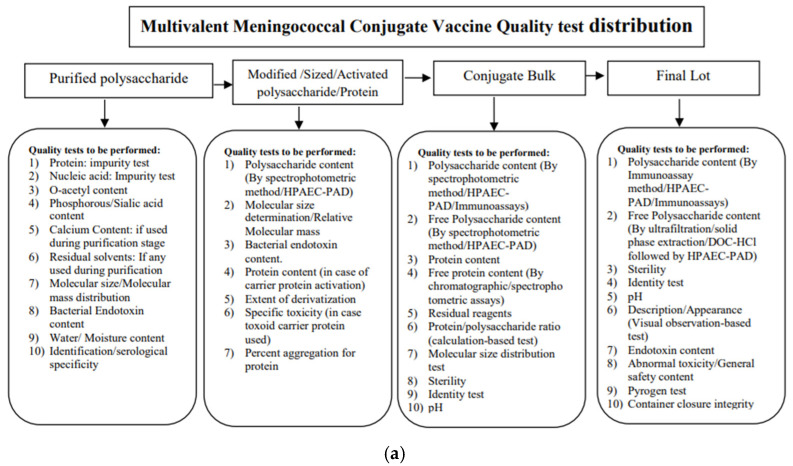
(**a**) Multivalent meningococcal conjugate vaccine quality control test distribution; (**b**) Chromatographic methods and their classification for multivalent meningococcal vaccine testing.

**Table 1 vaccines-12-01227-t001:** WHO prequalified Meningococcal Vaccines.

Serogroup	Manufacturer	Type	WHO PQ	Formulation	Shelf Life
A	Serum Institute of India	Conjugate	2010	Lyophilized	36 Months
ACYW	GlaxsoSmithKline (Italy)	Conjugate	2013	Lyophilized	36 Months
ACYW	Pfizer (United Kingdom)	Conjugate	2016	Lyophilized	36 Months
ACYW	Sanofi Pasteur (France)	Conjugate	2017	Liquid	24 Months
ACYWX	Serum Institute of India Pvt Ltd.	Conjugate	2023	Lyophilized (Free-dried)	36 Months

**Table 2 vaccines-12-01227-t002:** Meningococcal serogroup capsular polysaccharides, repeating units and characteristic monosaccharides.

Serogroup	Polysaccharide	Repeating Unit	Characteristic Molecule (After Hydrolysis)
A	Poly-α 1,6-N-acetylmannosamine-6-phosphate (variable C3 and C4 O-acetylation)	α 1,6-N-acetylmannosamine-6-phosphate monosaccharide	mannosamine-6-phosphate
C	Poly- α 2,9-N-acetylneuraminic acid (variable C7 and C8 O-acetylation)	α 2,9-N- acetylneuraminic acid monosaccharide	neuraminic acid
Y	Poly-α2,6-glucose-α1,4-N-acetylneuraminic acid (variable C7 and C9 O-acetylation)	α2,6-glucose-α1,4-N-acetylneuraminic acid disaccharide	glucose
W	Poly- α 2,6-galactose-α1,4-N-acetylneuraminic acid (variable C7 and C9 O-acetylation)	α 2,6-galactose-α1,4-N-acetylneuraminic acid disaccharide	galactose
X	Poly α 1,4 linked N-acetyl-d-glucosamine 1-phosphate	N-acetyl glucosamine 4-phosphate residues held together by α1-4 phosphodiester bonds.	glucosamine 4-phosphate

**Table 3 vaccines-12-01227-t003:** Standards used for Multivalent meningococcal vaccines.

Method/Test	Spectrophotometric/Chromatographic	Standard Used
Protein content	Spectrophotometric (Lowry/BCA)	Bovine serum albumin
Nucleic acid content	Spectrophotometric (Absorption at 260 nm)	Salmon DNA
Phosphorous content	Spectrophotometric (Chen’s method)	Phosphorous solution, Potassium dihydrogen phosphate, Trisodium phosphate dodecahydrate,
IPC-MS	Mass spectrophotometer	Phosphorous IPC MS standard
Sialic acid content	Spectrophotometric (Sevenerholm’s method)	N-acetylneuraminic acid
O-acetyl content	Spectrophotometric (Hestrin’s method)	Acetylcholine chloride
Endotoxin content	Spectrophotometric (LAL test)	Control Standard Endotoxin
Molecular size distributions/Purity by HPLC	Chromatographic (Size exclusion)	Salmon DNA, Thyroglobulin, Carbonic anhydrase, Ethylene glycol, 4-amino benzoic acid, Tyrosine, BSA etc,
Polysaccharide content	Chromatographic (Ion chromatography)	1st WHO International Standard for Meningococcal Group A (13/246) 1st WHO International Standard for Meningococcal Group C (08/214) 1st WHO International Standard for Meningococcal Group X (14/156) 1st WHO International Standard for Meningococcal Group Y (16/206) 1st WHO International Standard for Meningococcal Group W (16/152) or In house polysaccharide from serogroups A, C, Y, W and X
Residual Content	Chromatographic/ Spectrophotometric	Calcium/Acetonitrile/Ethanol/Adipic dihydrazide/1-ethyl-3-(3-dimethylaminopropyl) carbodiimide hydrochloride (EDC) content or any other residual reagent as applicable

**Table 4 vaccines-12-01227-t004:** Potential system suitability for non-chromatographic methods.

Test	Applicable Serogroups	Assay/Method	System Suitability Parameter	Applicable
O-acetyl content	A, C, Y, W	Hestrin’s Method	Correlation	*√*
			Slope	*√*
			% Recovery of standards	*√*
			% CV of duplicate std/Samples	*√*
			Internal Control	×
			Reference Standard	×
			International Standard	×
Phosphorous content	A and X	Chen’s Method	Correlation	*√*
			Slope	*√*
			% Recovery of standards	*√*
			% CV of duplicate std/Samples	*√*
			Internal Control	*
			Reference Standard/QC Control	*√*
			International Standard	*
Sialic acid Content	C, Y, W	Sevenerholm’s Method	Correlation	*√*
			Slope	*√*
			% Recovery of standards	*√*
			% CV of duplicate std/Samples	*√*
			Internal Control	*
			Reference Standard/QC Control	*√*
			International Standard	*
Protein content	A, C, Y, W, X	Lowry/BCA Method	Correlation	*√*
			Slope	*√*
			% Recovery of standards	*√*
			% CV of duplicate std/Samples	*√*
			Internal Control	*
			Reference Standard/QC Control	*√*
			International Standard	×

* Conditionally applicable.

**Table 5 vaccines-12-01227-t005:** Potential system suitability for chromatographic methods.

Test	Applicable Serogroups/Component	Assay/Method	System Suitability Parameter	Applicable
Total PS & Free PS content (Drug substance stage and Drug product stage)	A, C, Y, W and X	DOC-HCl* & HPAEC-PAD * for free ps estimation	Correlation	*√*
			Slope	*
			% Recovery of standards	*
			Internal Control	*√*
			Reference Standard/QC Control	*
			International standard	*√*
Purity by High performance size exclusion chromatography for Tetanus toxoid	Tetanus toxoid	HPLC	Correlation	×
			Slope	×
			Internal Control	*√*
			Reference Standard/QC Control	*
			International standard	×
			Resolution	*√*
			Plate Count	*√*
			Symmetry	*√*
Molecular size distribution in NmCV-5 purified polysaccharides	A, C, Y, W and X	HPLC	Correlation	*
			Slope	*
			Internal Control	*√*
			Reference Standard/QC Control	×
			International standard	×
			Resolution	*√*
			Plate Count	*√*
			Symmetry	*√*
Estimation of Free Protein in drug substance	A, C, Y, W and X	HPLC	Correlation	*√*
			Slope	*
			Internal Control	*√*
			Reference Standard/QC Control	×
			International standard	×
			Resolution	*√*
			Plate Count	*√*
			Symmetry	*√*

* Conditionally applicable.

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
