# Peer review of "Analytical Challenges in Novel Pentavalent Meningococcal Conjugate Vaccine (A, C, Y, W, X)"

_vaccines, 2024, doi:10.3390/vaccines12111227_

Round 1

Reviewer 1 Report

Comments and Suggestions for Authors

The authors reviewed the literature on “Analytical Challenges In Novel Pentavalent Meningococcal Conjugate Vaccine (A, C, Y, W, X)” for a better understanding of the advantages and disadvantages of different analytical methods for analyzing the Meningococcal vaccines.

 Abstract:

Well written and informative.

Line 18 and 27: what do CNB and IND stand for?

Introduction:

Good literature review but please don't forget to use a complete form of sentences or terms, the first time, before using an abbreviated form of them.

Please check the title of each paragraph. You jumped from the introduction to “ 2.0. Neisseria meningitides and Vaccine development” Where is one?

Lines78-79: It is not easy to follow.

Line 108 and lines after it: please use the standard scientific abbreviation style. “N. Meningitidis”

Line 168: In Figure 3, please mark for readers which branch of the flowchart is for a new vaccine and which one is for an already established vaccine. 

Comments on the Quality of English Language

The article needs a modrate revision of the writing format. 

Author Response

Reviewer 1:

Comment: Line 18 and 27: what do CNB and IND stand for?

Response: Authors would like to thank reviewer for valuable feedback and review comments on our manuscript. CNBr represents the cyanogen bromide based activations whereas, IND stands for Investigational new drug. Long forms are provided in abbreviations sections in manuscript file

Introduction:

Comment: Good literature review but please don't forget to use a complete form of sentences or terms, the first time, before using an abbreviated form of them.

Response: Corrected as per suggestions of reviewer.

Comment: Please check the title of each paragraph. You jumped from the introduction to “ 2.0. Neisseria meningitides and Vaccine development” Where is one?

Response: Introduction sections was highlighted as sections 1.0 in the world file uploaded in journal portal, however, in file layout for review it was not assigned 1.0.

Comment: Lines78-79: It is not easy to follow.

Response: Corrected as per suggestions of reviewer. The sentence has been rephrased for clarity.

Comment: Line 108 and lines after it: please use the standard scientific abbreviation style. “N. Meningitidis”

Response: Corrected as per suggestions of reviewer.

Comment: Line 168: In Figure 3, please mark for readers which branch of the flowchart is for a new vaccine and which one is for an already established vaccine. 

Response: Corrected as per suggestions of reviewer.

Comment: Comment on English quality

The article needs a moderate revision of the writing format.

Response: Authors would like to thank reviewer for valuable feedback and review comments on our manuscript. As per suggestions manuscript is revised for better understanding and readability.

Reviewer 2 Report

Comments and Suggestions for Authors

Sharma et al. aim to review current vaccines against meningococcal serogroups and highlight challenges, as well as knowledge gaps, in multivalent conjugate vaccine development. This topic is of interest to readers, particularly as the World Health Organization’s (WHO) roadmap sets a vision for 2030: "Towards a world free of meningitis." However, in its current form, this review does not meet the quality, or standards expected of such work. Additionally, the text is extremely difficult to understand, requiring rewriting and restructuring.

Please clarify what the authors mean by the following statement:  “ This obviously invite challenges while set- ting up appropriate quality control analytical tools..”

Please clarify what the authors mean by “We could able provide an approach based on challenges encountered and conclude on key areas of analytics viz. potency estimation and their challenges in complex multivalent vaccines, free saccharides and free protein  estimation and their potential gaps and challenges, implementation challenges for MSD to understand lot to lot consistency in multivalent conjugate vaccines.” 

What is meant by: “Figure 3. Process flow challenges in new vaccine analytics vs established vaccine analytics.”

Additionally, prepositions seem to be missing from figure titles. Please correct this: “Figure 2. Factors accountable analytical challenges.” to "Figure 2. Factors accountable for analytical challenges."

There are numerous examples of grammatical errors, poorly constructed sentences, and overall poor English throughout the manuscript. Additionally, some sentences are excessively long and lack meaningful information. This is a significant concern, and the quality of English must be improved throughout to meet basic and appropriate standards.

Furthermore, the figures and graphical illustrations appear basic and could be enhanced for the benefit of the readers.

Minor comments:

Line 18: There is a typo – please check the formatting of “various conjugation chemistries.”

Line 26-27: Please rephrase "This article presents an understanding" to clarify what is meant.

Line 27-29: The sentence "It highlights key challenges encountered and approaches exercised to address the problem through development of strategic analytical methods for key parameters" is unclear and needs rephrasing for clarity.

Line 14: There is a typo – please correct.

Page 2; lines 49-56: Please check and correct the font size.

Comments on the Quality of English Language

There are numerous examples of grammatical errors, poorly constructed sentences, and overall poor English throughout the manuscript. Additionally, some sentences are excessively long and lack meaningful information. This is a significant concern, and the quality of English must be improved throughout to meet basic and appropriate standards.

Author Response

Comment: Sharma et al. aim to review current vaccines against meningococcal serogroups and highlight challenges, as well as knowledge gaps, in multivalent conjugate vaccine development. This topic is of interest to readers, particularly as the World Health Organization’s (WHO) roadmap sets a vision for 2030: "Towards a world free of meningitis." However, in its current form, this review does not meet the quality, or standards expected of such work. Additionally, the text is extremely difficult to understand, requiring rewriting and restructuring.

Response: Authors would like to thank reviewer for valuable feedback and review comments on our manuscript. As per suggestions manuscript is revised for better understanding and readability.

Comment: Please clarify what the authors mean by the following statement:  “ This obviously invite challenges while set- ting up appropriate quality control analytical tools..”

Response: The sentence has been reconstructed in previous context for better understanding asHence, creating the appropriate quality control analytical tools is a challenge.”

Comment: Please clarify what the authors mean by “We could able provide an approach based on challenges encountered and conclude on key areas of analytics viz. potency estimation and their challenges in complex multivalent vaccines, free saccharides and free protein  estimation and their potential gaps and challenges, implementation challenges for MSD to understand lot to lot consistency in multivalent conjugate vaccines.” 

Response: The sentence has been rephrased for clarity as per suggestions of reviewer

Comment: What is meant by: “Figure 3. Process flow challenges in new vaccine analytics vs established vaccine analytics.”

 Response: Graphical representation attempted to give insight analytics challenges in established products vs products under development. Figure 3 and figure caption is revised for better clarity and understanding.

Comment:  Additionally, prepositions seem to be missing from figure titles. Please correct this: “Figure 2. Factors accountable analytical challenges.” to "Figure 2. Factors accountable for analytical challenges."

Response: Corrected as per suggestions of reviewer

Comment: There are numerous examples of grammatical errors, poorly constructed sentences, and overall poor English throughout the manuscript. Additionally, some sentences are excessively long and lack meaningful information. This is a significant concern, and the quality of English must be improved throughout to meet basic and appropriate standards.

Response: As per suggestions entire manuscript is revised for better understanding and readability.

Comment: Furthermore, the figures and graphical illustrations appear basic and could be enhanced for the benefit of the readers.

Response: Corrected as per suggestions of reviewer

Minor comments:

Comment:  Line 18: There is a typo – please check the formatting of “various conjugation chemistries.”

Response: Corrected as per suggestions of reviewer. The sentence has been rephrased for clarity.

Comment:  Line 26-27: Please rephrase "This article presents an understanding" to clarify what is meant.

Response: Corrected as per suggestions of reviewer. The sentence has been rephrased for clarity.

Comment:  Line 27-29: The sentence "It highlights key challenges encountered and approaches exercised to address the problem through development of strategic analytical methods for key parameters" is unclear and needs rephrasing for clarity.

Response: Corrected as per suggestions of reviewer. The sentence has been rephrased for clarity.

Comment: Line 14: There is a typo – please correct.

Response: Corrected as per suggestions of reviewer.

Comment: Page 2; lines 49-56: Please check and correct the font size.

Response: Corrected as per suggestions of reviewer.

Comments on the Quality of English Language

Comment:  There are numerous examples of grammatical errors, poorly constructed sentences, and overall poor English throughout the manuscript. Additionally, some sentences are excessively long and lack meaningful information. This is a significant concern, and the quality of English must be improved throughout to meet basic and appropriate standards.

Response: As per suggestions entire manuscript is revised for better understanding and readability.

Round 2

Reviewer 2 Report

Comments and Suggestions for Authors

The revised version does not meet the journal's required standard.

Below are a few early examples, but there are numerous similar issues throughout the manuscript that the authors need to address.

 1.       All figures use inconsistent (random) fonts and sizes.

2.       The wording is awkward/ clunky, with an overuse of “the.”

3.       The final sentence of the abstract is vague and should include more details about the discussion points.

4.       Phrases like “Having said that” should be revised.

5.       Lines 6-9 in the introduction are poorly written.

6.       Abbreviations such as “MSD” are not defined.

7.       "N. meningitidis is known as an" should be "N. meningitidis is a…"

8.       "In the year 2010" should be "In 2010."

9.       Figure 1 is poorly presented and lacks substantial information.

10.   Figure 2 contains typos, such as "multiple different carrier proteins," which should be "multiple carrier proteins."

11.   Figure 3 has a typo: "license product" should be "licensed product."

12.   "Analytics Challenges and limitations" should be "Analytical Challenges and limitations."

Comments on the Quality of English Language

The English is very difficult to understand/incomprehensible.

Author Response

The revised version does not meet the journal's required standard.

Below are a few early examples, but there are numerous similar issues throughout the manuscript that the authors need to address.

  1. All figures use inconsistent (random) fonts and sizes.

Response: Corrected as per suggestions, all the figures are harmonized to one font, however font size varied due to presentation limitations but is ensured to keep it either 14 or 12.

  1. The wording is awkward/ clunky, with an overuse of “the.”

Response: Entire manuscript is checked and corrected for language and grammar as suggested by reviewer.

  1. The final sentence of the abstract is vague and should include more details about the discussion points.

Response: As per suggestions for better clarity the sentence is revised

  1. Phrases like “Having said that” should be revised.

Response: Sentence is removed from the manuscript.

  1. Lines 6-9 in the introduction are poorly written.

Response: As per suggestions of the reviewers Line 6 to 9 are rephrased for better clarity.

  1. Abbreviations such as “MSD” are not defined.

Response: Abbreviations MSD is described in abbreviations section, also is written in full form at its first appearance as per suggestions of the reviewer

  1. "N. meningitidis is known as an" should be "N. meningitidis is a…"

      Response: Sentence corrected as per suggestions removed word exclusive

  1. "In the year 2010" should be "In 2010."

      Response: Corrected as per suggestions

  1. Figure 1 is poorly presented and lacks substantial information.

      Response: Figure 1 is removed in the context of the manuscript flow.

  1. Figure 2 contains typos, such as "multiple different carrier proteins," which should be "multiple carrier proteins."

      Response: Figure 2 is corrected as per suggestion , since figure 1 is removed now it is renumbered as figure 1

  1. Figure 3 has a typo: "license product" should be "licensed product."

      Response: Figure 3 is corrected as per suggestion , since figure 1 is removed now Figure 3  is renumbered as figure 2

  1. "Analytics Challenges and limitations" should be "Analytical Challenges and limitations."

      Response: Corrected as per suggestions

The English is very difficult to understand/incomprehensible

Response: Entire manuscript is checked and corrected for language and grammar as suggested by reviewer.

Round 3

Reviewer 2 Report

Comments and Suggestions for Authors

Comments and Suggestions for Authors:

While the authors have addressed some of the concerns raised, I believe further improvements can be made, and extensive editing of the English language is required. I recommend that the authors consider using MDPI's English Language Editing Services or a similar service. This will enhance the quality of the manuscript for the benefit of the readers.

Comments on the Quality of English Language

 Extensive editing of English language required.